

# Proximate and underlying drivers of socio-hydrologic change in the upper Arkavathy watershed, India

Veena Srinivasan[1], Gopal Penny[2], Sharachchandra Lele[1], Bejoy K. Thomas[1], and Sally Thompson[2]

[1]Ashoka Trust for Research in Ecology and the Environment, Royal Enclave, Sriramapura, Bengaluru, KA, India
[2]Department of Civil and Environmental Engineering, University of California, Berkeley, CA, USA

*Correspondence to:* Veena Srinivasan (Veena.srinivasan@atree.org)

**Abstract.** Addressing water security in the developing world involves predicting water availability under unprecedented rates of population and economic growth. Yet the combination of rapid change, inadequate data and human modifications to watersheds poses a challenge, as researchers face a poorly constrained water resources prediction problem. This case study of the data-scarce,

upper Arkavathy watershed, near the city of Bengaluru in southern India, attempts to systematically explain the observed disappearance of surface and groundwater in recent decades. The study asks three questions: 1) Can we quantify the change attributable to different drivers? 2) Can we anticipate change? 3) What policy lessons can be drawn? Field experiments, isotopic studies, borewell scans and sensors were deployed to understand hydrologic processes over five years. These were used in a

historical reconstruction of the catchment over three decades that replicates the decline. The multi-scale model of the upper Arkavathy, quantifies the contributions of soil and water conservation measures, groundwater depletion and eucalyptus plantations to the decline in surface and groundwater resources. The model results indicate that the catchment hydrology cannot be reconstructed without explicitly including human feedbacks. The system is influenced by both endogenous drivers (so-

cial feedbacks to changes in water availability such as irrigation efficiency improvements, soil and water conservation measures and deeper borewells) and exogenous drivers (technological change, pro-development governance and economic forces due to urbanization, which provided access to capital and markets for high-value crops). The research suggests that in a system where productivity of the landscape is limited by water, economic drivers will always push for maximization of water

abstraction and use. Unsustainability of resource use is inevitable, in the absence of credible controls on abstraction and use,




## 1 Introduction

As populations grow, humanity faces the prospect of uncertain future water supplies both due to climate change and increasing demands on water (Wagener et al., 2010). There is considerable interest
in the global community in tackling the threat of water insecurity (Cook and Bakker, 2012). Predicting the future of water availability and water demand at regional scales is central to the endeavor of tackling water insecurity (Falkenmark and Rockström, 2006).

Yet this prediction goal is complicated by the acknowledged challenge of non-stationarity (Milly et al., 2008), which renders many conventional prediction tools inapplicable, even on decadal timescales
(Srinivasan et al., 2017). In large part, hydrologic non-stationarity on short to medium timescales derives from direct anthropogenic influences on the water cycle, which may be much more significant than the more widely studied impacts of climate change (Kumar et al., 2006; Vorosmarty, 2000; Taylor et al., 2013; Schewe et al., 2014). Anthropogenic drivers of non-stationarity include changes in land use (Fox et al., 2015), direct manipulation of surface water resources (Grafton et al., 2013)
and groundwater abstraction (Zeng and Cai, 2014).

The problems of prediction under change are compounded in the developing world, which is experiencing unprecedented rates of population, economic and infrastructural growth. The net impact is *rapid* environmental change, occurring in the absence of long-term hydrologic records, and frequently the absence of local hydrologic process understanding. Moreover, many regions experi-
encing rapid are characterized by hydrologic conditions such as hard-rock aquifers, low catchment storage, high aridity and high variability, which tend to amplify the impacts of these changes. Poor and vulnerable communities bear the greatest burden.

Urgent action to address the prediction problem and guide water development approaches worldwide is clearly needed. Yet, despite recent efforts (Mondal et al., 2017) the scientific basis for robust
decision making around water in the developing world is still absent (Srinivasan, 2017). The combination of rapid change, water scarcity, inadequate data, human feedbacks, and urgent need for action present a thorny problem for researchers and decision makers alike. Research in the developing world is highly pertinent because much of the infrastructure is still being built and demand for water is still sharply rising; decisions made today may shape water security in these regions for the
next 100 years.

Typically, hydrologists have focused on the direct effects of anthropogenic forcing factors by positing a one-to-one relationship between a change in water or land use and an associated hydrologic response. While there has been a strong focus on *proximate* drivers of change (e.g. a change in land use), research on *underlying* anthropogenic drivers (e.g. why land use changed) is still nascent.
Often, the unstated assumption is that the underlying structure and function of both social and physical systems remain stationary. Yet emerging case studies demonstrate that anthropogenic drivers can result in hydrologic regime shifts - for example threshold-like changes in runoff ratio following groundwater disconnection from surface networks (Kinal and Stoneman, 2012). Similar punctuated





changes can be expected in underlying social drivers, for example those associated with changes
in economics, technology and governance. Most hydrologic models are not equipped to diagnose
such changes in historical records, let alone anticipate adaptive responses of different actors to fu-
ture infrastructural, social or policy changes. The current IAHS decade starting 2013, "Panta Rhei"
(Montanari et al., 2013; McMillan et al., 2016), reflects these realities, and identifies that the frontier
of hydrologic prediction involves predictions under change (Thompson et al., 2013; Srinivasan et al.,
2017). The discipline of socio-hydrology is also emerging as a science that explicitly accounts for
feedbacks between water and society (Sivapalan and Blöschl, 2015; Sivapalan et al., 2012).

Yet primary research that systematically unbundles the causes of hydrologic change in rapidly
developing regions, with specific attention to underlying human drivers, remains rare. This study
of the upper Arkavathy (TG Halli) Watershed near Bengaluru (Bangalore) in the state of Karnataka,
southern India is an attempt to understand the nature and causes of rapid hydrologic change in a data-
scarce region, and interpret four decades of physical change in terms of proximate human drivers,
and their underlying causes.

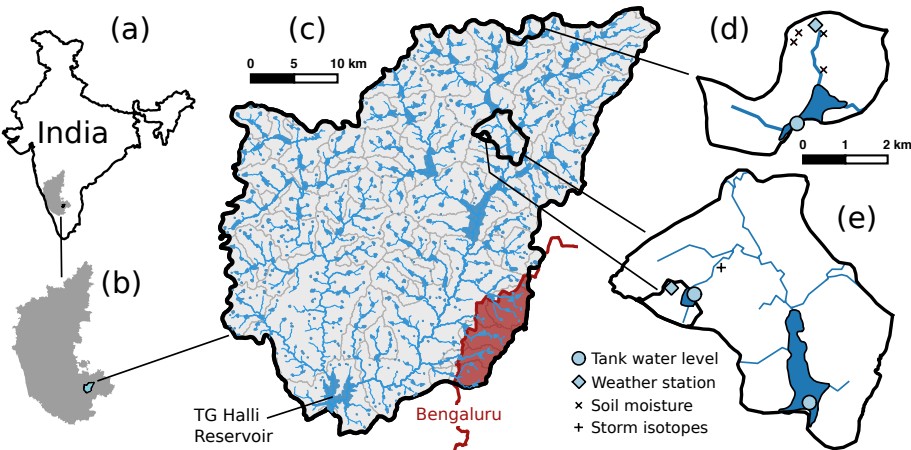

**Figure 1.** Map of the upper Arkavathy Watershed, showing regional context within (a) India and (b) Karnataka
state, (c) the TG Halli watershed, and (d,e) two intensively-studied "milliwatersheds". The Bengaluru (Banga-
lore) urban area is shown in red on the eastern boundary of the watershed.

## 2 Study Area

The upper Arkavathy (TG Halli) watershed lies to the west of the rapidly growing metropolis of
Bengaluru city (Bangalore), one of India's fastest growing metropolitan areas (Fig. 1). The area
is seasonally monsoonal, receiving about 800 mm of precipitation annually, both from the south-



west (Jun-Sep) and north east (Oct-Dec) monsoons. The main stem of the Arkavathy River has its headwaters in the Nandi Hills north of Bengaluru and is joined by its first major tributary, the Kumudavathy River at Thippagondanahalli village, where the Thippagondanahalli (TG Halli) reservoir was constructed in 1935 to supply water to Bengaluru. The TG Halli resevoir defines the terminus of the upper Arkavathy watershed, with a total drainage area of 1447 sq. km (Lele et al., 2013).

The watershed contains an older water supply reservoir called Hesaraghatta as well as an estimated 617 "tanks", small surface water storage structures with a cumulative capacity of 143 Million cubic metres (MCM); 1.5 times the storage of TG Halli reservoir. The tanks historically captured a proportion of the surface flow. Overflows from upstream tanks would feed those downstream in a cascade (Vaidyanathan, 2001; Shah, 2003). Farmers used the tank water to irrigate a second winter crop after the monsoons. Tanks were operated by village water-men or "neeragantis" who received a tithe of crop production as payment for their efforts, integrating agriculture and water management in village social structures. The tanks rarely provided inter-annual storage.

The watershed is underlain by low-yielding "hard-rock" aquifers (Das, 2011). The aquifers consist of an unconsolidated weathered mantle with varying thickness (on the order of meters), and a more extensive (order tens of meters) fractured-weathered layer, underlain by unweathered bedrock (Maréchal et al., 2004). Fracture densities generally decline with depth (Maréchal et al., 2004).

Over the past four decades, inflows into TG Halli reservoir have declined by almost 90 percent (Srinivasan et al., 2015). Over the same period, other tanks in the watershed also exhibited a drying trend (Penny et al., 2017), and groundwater levels dropped by hundreds of metres. Analysis of historical rainfall from four rain gauge locations within the watershed suggests that neither annual average rainfall nor daily rainfall volumes (taken as a proxy for storm intensity) have changed significantly. Potential evapotranspiration (PET) inferred from long term temperature records did not change significantly either, and no new large dams were built.

However, there were several potentially important anthropogenic changes in the watershed over the period of interest. Eucalyptus plantation areas expanded ten-fold, thousands of small rainwater retention and harvesting structures were constructed, and private farm borewells were increasingly installed on private landholdings (Srinivasan et al., 2015). These changes are hypothesized to be the proximate anthropogenic drivers of the observed drying.

This study builds on the previous study to answer three questions:

1. Can the change attributable to each of the proximate drivers be quantified?

2. Can we anticipate change? Will understanding these underlying drivers and their role improve predictive insights?

3. What are the implications of the answers to these questions for improving management of the upper Arkavathy watershed?





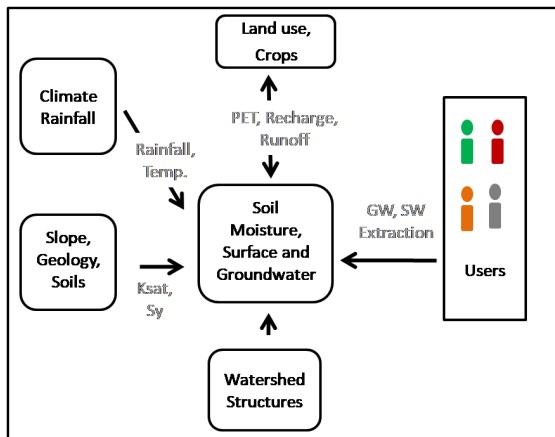

**Figure 2.** Simple conceptual model of the hydrology of TG Halli watershed.

This study aimed to answer these questions, by leveraging five years of research involving nearly 50 scientists in the watershed. This effort enables us to at partially overcome the challenges posed by data scarcity and develop a narrative of coupled social and physical changes over the past four decades.

## 3   Research approach

To synthesize the dominant hydrologic processes in the upper Arkavathy watershed we (i) identified the primary (proximate) drivers controlling the socio-hydrologic system of the TG Halli watershed, (ii) assembled datasets that describe these drivers and their evolution over the past four decades, and (iii) conducted a model-based (or "theory-based") reconstruction to reproduce the extensive changes in the socio-hydrologic system that occurred a four-decade period from 1975 to 2015. Reconstructing the hydrology of the upper Arkavathy is complicated by the limited historical records in the region (environmental or social) and the decentralized and uncoordinated actions of millions of people. Simulating the history of this watershed without sufficient understanding of the dynamics of the system could produce a model that reproduces the appropriate outcomes using the wrong dynamics (Beven, 2011), in part because human-induced non-stationarity can introduce parameter and model structural errors over time (Beven, 2006).

One approach to minimize the problem of equifinality is to conduct a process-based reconstruction to understand the mechanistic connections between drivers and outcomes within the watershed, using it as a precursor to theory-based modeling (Penny et al.).

A conceptual model of the hydrology of the upper Arkavathy watershed was proposed, linking physical response with proximate anthropogenic change drivers (Fig. 2). This conceptual model




framed the scope of the extensive field research campaign to collect data that represents the important system parameters and processes (Section 4). The data were then used to simulate the long-term

dynamics of the watershed in a hydrologic model (Section 5).

## 4   Data

Data were collected for each of the drivers in the box shown in Figure 2. A wide variety of techniques were used - field experiments, isotopic studies, borewell scans, remote and in-situ sensing to characterize hydrologic processes, and well censuses, surveys, policy-document analyses, focus

group discussions and interviews to characterize social processes. The majority of this research was undertaken within two focus study sites, termed "milli-watersheds" (Fig. 1) with representative characteristics of the upper Arkavathy watershed as a whole in terms of agricultural land use and irrigation management. Key findings from these efforts, and the relevant data inputs to the hydrology model, are outlined in the following sections.

### 4.1   Climate and physiography

Daily rainfall data were obtained for four rain gages in the upper Arkavathy watershed from 1970-2014 from the Karnataka Department of Economics and Statistics (DES, 1970). 15-min data from 2010-2015 were made available for three rain gages by the Karnataka State Natural Disaster Monitoring Cell (KSNMDC). Weather stations were installed in each of the two milli-watersheds in 2014

(see Fig 3 for a watershed-averaged timeseries of rainfall). Daily weather parameters were obtained for Bangalore Rural District from the Indian Meterological Department (IMD) to estimate potential evaporation. The daily rainfall and weather data from 1970 were statistically downscaled using 30-min data from KSNMDC and the ATREE weather stations to provide model input.

    A digital elevation model using 30m SRTM data along with survey of India topographic sheets

published in the 1970s, were used map tanks, streams and sub-watershed boundaries. Overflow from each tank is routed to the next tank downstream. Tank sizes in the watershed ranged from 0.1 to 8.7 $km^2$.

### 4.2   Soil Moisture and Surface water

Inflows into TG Halli reservoir were reported by Bangalore Water Supply and Sewerage Board

(BWSSB) which owns and operates the reservoir. Daily inflows were reported from 1977, and monthly inflow from 1937.

    Bathymetry of the sample tanks was established using a combination of visual imaging from drones (for dry tanks) and ultrasonic sensors on boats (for wet tanks) and used to obtain a general scaling relationship between stage, water-spread area and volume (Young et al., 2017).





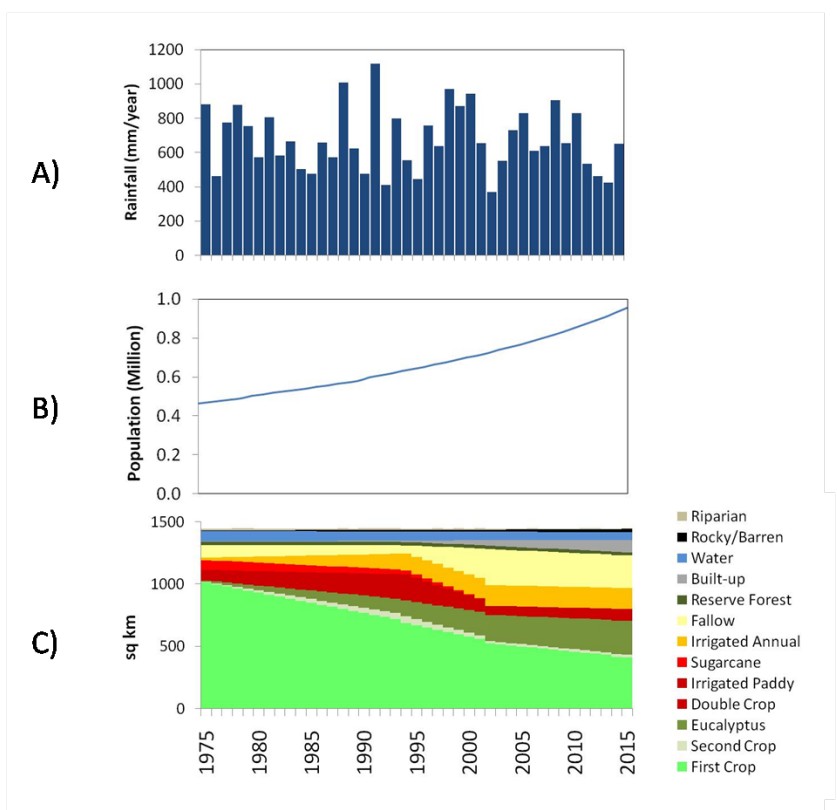

**Figure 3.** Examples of time variation in key watershed characteristics A) Rainfall, B) Population, C) Land use

Soil surface saturated hydraulic conductivity ($K_{Sat}$) was measured using infiltrometry in 60 plots with different land uses; soil moisture and isotopic studies were conducted to understand the dominant runoff generation processes (Penny et al.). Soil profile data and soil maps were obtained from the National Bureau of Soil Survey and Land Use Planning (NBSSLUP), and showed that soil type was largely uniform across the watershed.

**4.3   Groundwater**

A geologic map was obtained from the Geological Survey of India, which showed the dominant formations to be granites and gneisses. In the absence of pump tests to understand the aquifer characteristics, borewell camera scans (Fig S3.5) were crowd-sourced from local commercial operators for ≈150 borewells. The scanned wells were geo-tagged and the depth and status (wet or dry) of sub-
surface fractures were recorded. The borewell camera scans were used to estimate the bulk porosity or specific yield of the fractured rock aquifer (Fig S3.8).



The field team conducted an open well census and a borewell census in the two milli-watersheds. Every open and bore well (abandoned or in use) was geo-tagged and well owners interviewed to identify depths at which farmers had encountered water over time. The groundwater history obtained

from the well census was used reconstruct the rate of groundwater depletion over time (Fig 5). It allowed the project team to establish that long-term monitoring wells in the upper Arkavathy were not representative of the trends in the majority of the watershed, therefore data from monitoring wells were discarded.

### 4.4 Crop choice and land use

Secondary data at the sub-district level published in state Annual Season Crop Reports were collected from the Karnataka Department of Economics and Statistics (DES, 1970) from 1970 to 2012.

To understand how farmers make plot-level decisions, we used a primary dataset of farmers from 15 villages within the upper Arkavathy watershed. The villages were selected using a stratified random sample based on labor and water availability. Field research was conducted during 2013-2015

and included household level questionnaire surveys to collect information about crops cultivated by season, irrigation behavior, income and water management. This was supplemented by verification of crops and irrigation sources in agricultural plots, focus group discussions with farmers and interviews with key local informants including government officials (Patil et al., 2017).

Land use at the watershed scale was reconstructed by mapping historical land use using satellite

imagery (Table 1, Fig 3). Land use classes relevant to infiltration and evapotranspiration were selected. To validate the supervised classification (Maximum Likelihood classification with majority filter) 1250 ground truth points were collected (Lele and Sowmyashree, 2016).

### 4.5 Irrigation

The Annual Season Crop Report (ASCR) data show that tanks (67%) and open wells (33%) were the

only sources of irrigation in 1975. Today, 100% of the irrigation is from deep borewells (Fig S2.2, S2.4 and S2.6). Irrigation technology has improved from being entirely flood irrigation in the 1970s. A farm plot level census of *all* irrigated plots in the two milli-watersheds indicated that 48% and 68% of the irrigated area in Hadonahalli and Aralumallige respectively was under drip irrigation in 2016 (S2: Fig S2.9).

### 4.6 Watershed Structures

Runoff generated at the plot scale may re-infiltrate or evaporate after being captured by farm bunds (mud structures bordering individual farmers' fields) and/or check dams (small masonry structures constructed within stream channels). These structures trap runoff, boost infiltration and trap sediment. Field teams walked through the milli-watersheds recording the condition, location, height,

and width of check dams (Fig S4.1, S4.2). They also mapped stream locations manually, as these



**Table 1.** Land use change in the upper Arkavathy watershed. Source: (Lele and Sowmyashree, 2016).

| Land use class | 1972-73 | 1993-94 | 2000-01 | 2013-14 |
|---|---|---|---|---|
| Single Rainfed Crop | 990 | 722 | 541 | 436 |
| Irrigated Rice | 68 | 49 | 0 | 0 |
| Irrigated Crops | NA | 144 | 73 | 91 |
| Perennial Irrigated | 87 | 168 | 166 | 168 |
| Eucalyptus[a] | 0 | 122 | 211 | 270 |
| Fallow Land | 96 | 61 | 288 | 264 |
| Reserve Forest | 28 | 28 | 27 | 27 |
| Built up | 2 | 11 | 49 | 95 |
| Water | 78 | 69 | 65 | 66 |
| Barren/Rocky | 15 | 17 | 17 | 24 |
| Riparian/Shrub | 0 | 9 | 10 | 4 |
| Cloud/Shadow | 84 | 45 | 0 | 2 |

[a] Eucalyptus on private land only, includes area under acacia and silver oak.

locations have shifted over time as farmers have encroached on stream beds. The milli-watersheds were found to have a density of 1.5 check dams per km$^2$, ranging from 0.5 m to 2 m in height (Fig S4.3). Farm bunds were estimated from pictorial evidence to range from about 0.1 to 0.25 m in height (Fig S4.4).

Automatic water level monitoring instruments were installed in eight suitable check dams. The bathymetry of each check dam was mapped with a dumpy-level instrument to develop stage-volume and stage-area relationships (Fig S4.5, S4.6). The maximum check dam volumes ranged from 21 $m^3$ to 113 $m^3$. Using a simple water balance approach, inflow into check dams was partitioned between evaporation, infiltration and overflow. Between 11% and 32% of check dam inflows evapo-

rated (Jeremaih et al., 2014).

## 5   Model

A multi-scale hydrologic model was used to reconstruct observed dynamics of the upper Arkavathy watershed over the time period of the study (1975–2015). The model allowed us to hypothesize drivers of change that could not be well quantified from field research or other available data sources.

The human interventions that shaped the hydrologic trajectory of the upper Arkavathy watershed for the past 40 years include physical structures (check dams and farm bunds), changing irrigation technologies, and land use change. These interventions occur at different scales, due to the actions of different human agents. To accommodate this, we used a nested modeling approach (Fig. 4) that





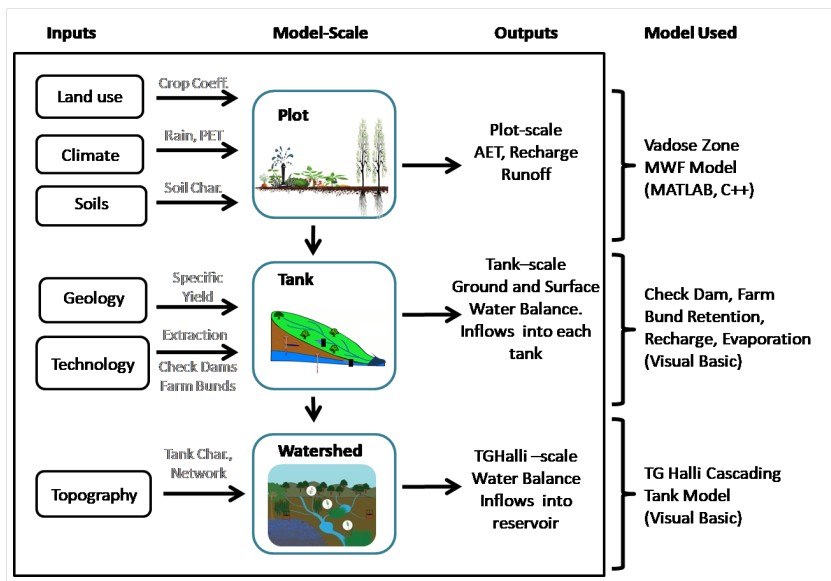

**Figure 4.** Nested modeling approach involved modeling hydrologic processes at the plot, tank and watershed scale

upscaled models from the plot to the tank and watershed scales (described below in Sections 5.1–
230  5.3).

The model equations are presented in more detail in Supplement S1. Model processes and parameters were prescribed using the findings and data from the field research campaign (Table S1: Parameter Table), except in cases where we were unable to collect adequate field data. For instance, saturated hydraulic conductivity and total capacity of farm bunding (the cumulative effects of soil
and water conservation practices) were treated as calibration parameters (see Section 5.4).

### 5.1  Plot-scale model

At the plot scale, farmers make land use, cropping and irrigation technology decisions that alter recharge, evapotranspiration, runoff and groundwater abstraction. The dominance of overland flow as the *contemporary* runoff generating mechanism in the watershed was established by a suite of
isotope and field analyses (Penny et al., 2017). Runoff was modeled at the plot-scale as a result of saturation excess and infiltration excess rainfall on 30-minute timescales using the multiple wetting front (MWF) model (Struthers et al., 2006). Rainfall, soil and rooting depth data were input into a 1-D water balance model which used potential evaporation from the Hargreaves Equation (Hargreaves and Allen, 2003) and FAO crop coefficients (Allan, 1998) (S1: Eq 1 and 2).





The MWF model was modified to accommodate possible groundwater mining by eucalyptus trees. Prior studies suggested that eucalyptus trees *can* transpire up to 2100 mm Yr$^{-1}$ (Calder et al., 1993), exceeding annual rainfall and thus implying such mining. As the groundwater table dropped in the MWF model, the fraction of eucalyptus trees mining groundwater was proportionately decreased.

### 5.2    Tank scale model

For each tank, a linked surface-groundwater model on a monthly time-scale was developed. 30-min runoff, recharge, evapotranspiration and groundwater abstraction from the MWF model were aggregated to the tank scale for each time period by multiplying by the area under each land use (S1: Eq 6-9).

The surface water in the tank watersheds were modeled as series of lumped storages accounting
for total storage in tanks, check dams and farm bunds. (S1: Eq 10-11). Based on field observations, it was assumed that all check dams were on first and second order streams, and not on streams connecting tanks.

The aquifer underlying each tank was similarly modeled as a simple lumped reservoir. Sub-surface connectivity between "tank aquifers" was assumed to be negligible as the borewell camera scans did
not show any evidence of large-scale fracture connectivity. In estimating the groundwater balance for each tank, recharge and extraction were based on land use from the MWF model.

The irrigation water requirement was assumed to be the difference between potential evapotranspiration and rainfed evapotranspiration (S1: Eq 3). Water abstracted was assumed to be exceed this requirement by an "irrigation sagacity" factor (Burt et al., 1997; Solomon and Burt, 1999) that
decreased with improved irrigation technology (S1: Eq 4). Irrigation water abstraction was split between ground and surface water based on the historically observed sources of irrigation (Fig. 5). Excess applied irrigation was assumed to evaporate.

If the groundwater level rose above the stream-bed (assumed to be 10 m below ground level), then the aquifer contributed to baseflow (S1: Eq 13). The difference between groundwater extraction and
recharge resulted in changing groundwater storage in each tank-aquifer (S1: Eq 14). Groundwater storage was divided by the specific yield to obtain the change in groundwater depth (S1: Eq 15).

### 5.3    Watershed-scale model

At the watershed scale, there is no continuous stream flow in the watershed, because of the cascading tank system. The upper Arkavathy watershed was modeled as a network of connected tanks (See Fig.
1), starting at the upstream tanks and going downstream.

If a tank filled up, any excess inflow would cascade to the next downstream tank (S1: Eq 18). Runoff from the whole watershed into TG Halli reservoir thus required continuous networks fo tanks to fill. In the absence of such continuously filled tanks, the only inflow into the reservoir is from its own sub-watershed of 49 $km^2$.





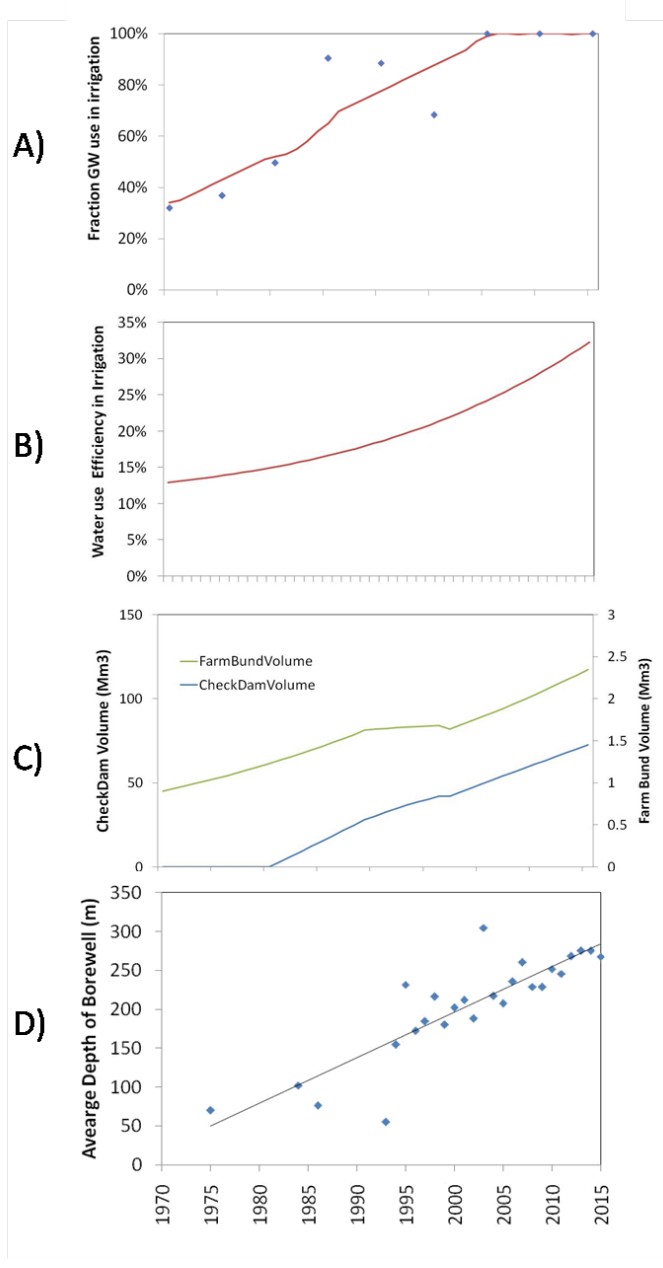

**Figure 5.** Endogenous Drivers: A) Irrigation water source B) Irrigation sagacity, C) Farm bunds and check dam volumes, D) Average borewell depths



### 5.4 Model calibration

Model calibration aimed to reproduce the observed monthly record of inflows to the TG Halli reservoir and depth to the groundwater table using $K_{Sat}$, irrigation sagacity, and farm bund coverage and heights as calibration parameters. These parameters were tuned so that simulations closely approximated observations.

One of the challenges in calibrating was that a sharp decline in runoff after 1995 could not be easily explained by climate, groundwater disconnection or land use change mechanisms. We hypothesized that this decline was due to the increase in decentralized storage created by soil and water conservation efforts (i.e., check dams and farm bunds). Of these, the volume of runoff retained in check dams was known from comprehensive field surveys (Fig. S4.1-S4.4).

Field bunds were observable visually but their extent could not be easily established, but rather they remained a calibrated parameter. It is difficult to untangle two cases - high $K_{Sat}$ with no field bunds (that results in no runoff generation even at the plot-scale) vs. a low $K_{Sat}$ with field bunds (that allows runoff generation at the plot-scale, but then traps it in the farm bund), but it is nevertheless possible constrain each. If the $K_{Sat}$/P ratio was set too high, then no runoff was generated at all, even in the 1970s. This scenario must be incorrect, because satellite images show tanks filling up in the 1970s (Penny et al., 2017). If the $K_{Sat}$/P ratio was set too low, too much runoff is produced to be contained behind field bunds with heights observed in the field; so intermediate values of $K_{Sat}$ and field bund heights were used.

The model was very sensitive to irrigation sagacity. Because the specific yield in the deeper fractures is barely 0.1 to 0.2% (Fig S3.9), small changes in net extraction resulted in big changes in groundwater levels. The specific yield parameter range was constrained both by published estimates and the borewell camera observations. Similarly, land use changes and cropping patterns over time are known, so there is *relatively* little uncertainty in crop water requirements. Irrigation water requirements alone cannot explain the high levels of groundwater abstraction needed to explain the declines. The *current* adoption of drip irrigation is known from field surveys; evaporative losses from inefficient irrigation had to be factored in the early years to explain the observed declines. If irrigation sagacity was increased too quickly in model runs, the early declines in groundwater could not be reproduced.

## 6 Results

The model was able to replicate inflows into TG Halli reservoir and depth to the groundwater table (Fig. 6), with the exception of TG Halli inflow in 1991-1995 and groundwater level in 2006-2010 (See S5 for further discussion). The simulation revealed the relative importance of different drivers in the upper Arkavathy watershed.



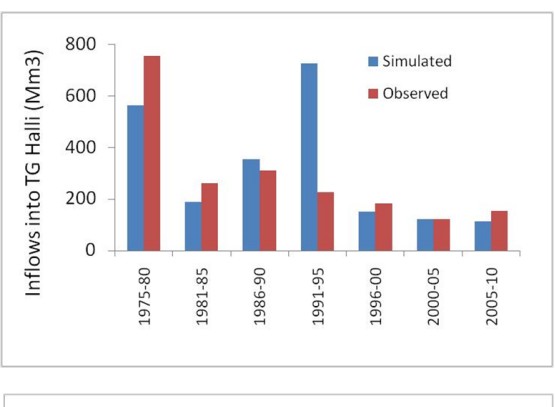

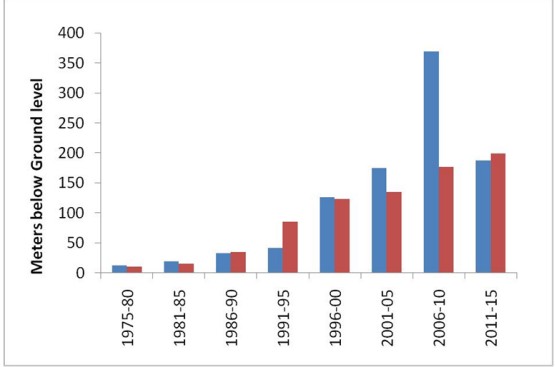

**Figure 6.** Comparison of simulated and observed surface inflows into TG Halli reservoir (upper panel) and groundwater levels inferred from well surveys (lower panel)

### 6.1 Plot-scale model results

At the plot scale, the MWF model results show that in land under shallow rooted rainfed crops, recharge is high and actual evapotranspiration is seasonal and relatively low (Fig. 7). In contrast, eucalyptus or sugarcane plantations generated virtually no recharge. AET for sugarcane and eucalyptus was much higher than fallow land or land under field crops. Because of the relatively high $K_{Sat}$/P ratio, runoff generation occurred only during high intensity events.

### 6.2 Tank-scale model results

The linked surface-groundwater model showed that most of the decline in runoff occurred due to the loss of baseflow prior to 1995. However, neither groundwater decline nor land use change could explain the loss of runoff after 1995. The dominant land use transformation observed in almost all the tanks was from rainfed cropland to built-up, fallow land or eucalyptus. None of these resulted in

an increase in $K_{Sat}$ (Penny et al., 2017; Penny et al.).



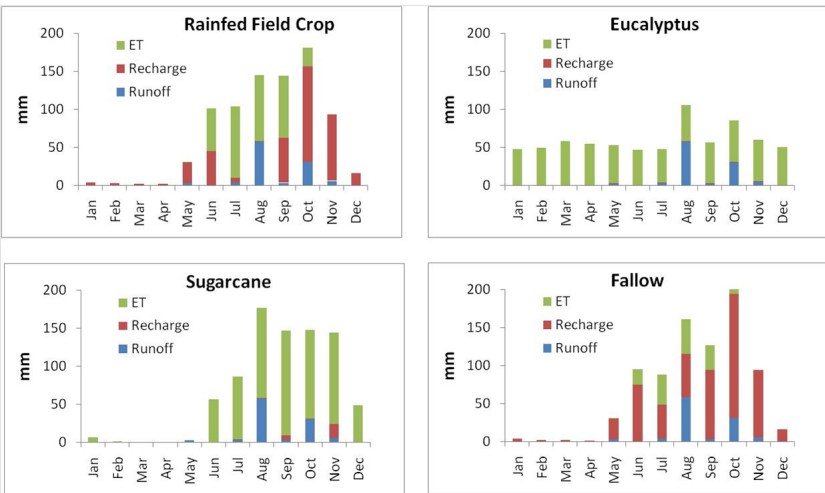

**Figure 7.** Outputs of plot-scale MWF Model under A) Rainfed Field Crop B) Eucalyptus C) Sugarcane and D) Fallow

Thus, an additional, non-stationary physical mechanism that could trap runoff and convert it into recharge was needed to explain continuing streamflow declines. Farm bunds retained a larger volume of runoff than did check dams because they are spread throughout the catchment area (Fig. 5), while the check dams are confined to the stream channels.

Overall, there was considerable heterogeneity in the trajectories of different tanks throughout the upper Arkavathy watershed. Sub-watersheds close to Bengaluru have urbanized much more rapidly, generated substantial sewage flows, flashier runoff and baseflow; they also had lower evapotranspirative losses as much of the land had been fallowed or paved.

### 6.3    Upper Arkavathy watershed-scale model results

At the scale of the whole upper Arkavathy watershed, from 1975 to 2002 there was a sharp increase in irrigated area and eucalyptus plantations. Between 1990 and 2002, modeled evapotranspiration exceeded average rainfall, suggesting that the most serious groundwater depletion occurred during this period. Post 2002, actual evapotranspiration (AET) in upper Arkavathy watershed was influenced by competing factors. On one hand, continuing expansion of eucalyptus increased AET. On

the other hand, irrigation water use efficiency improved and irrigated area began to drop. AET began to decrease after 2002 because of fallowing of land, though it remains higher than pre-development levels. Despite the increasing population, domestic water needs and sewage flows were not significant at the watershed scale even in 2015.

The anthropogenic drivers are not independent, making attribution challenging. The check dams

and farm bunds increase soil moisture and boost both evapotranspiration and recharge. Likewise




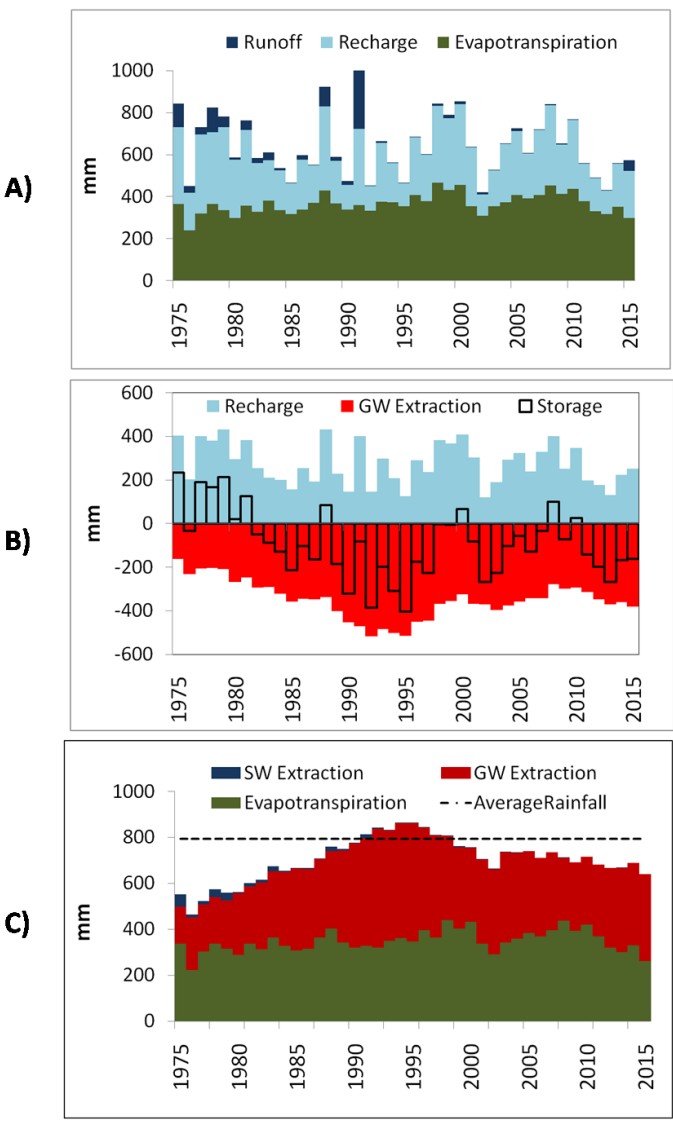

**Figure 8.** Outputs of the upper Arkavathy Watershed-scale Model. A) Partitioning of rainfall into runoff, ET and recharge, and B) Groundwater Balance. Note that here, "runoff" represents the total inflow into all tanks in the TG Halli Watershed, not inflows into TG Halli reservoir, which would be represented by inflows into the downstream most tank (Tank No 81).



**Table 2.** Attribution of ground water and surface water decline to different anthropogenic drivers

| Anthropogenic Drivers | Δ ET [a] |
|---|---|
| Eucalyptus Expansion | -13% (GW) |
| Fallow/Rainfed Crop Decline | +6% (GW) |
| Irrigated Agriculture Expansion | -5% (GW) |
| Soil and Water Conservation Measures | -4% (SW), +4% (GW) |
| Drip Irrigation Adoption | 10-25% (GW) |

[a] Comparing normalized figures for two similar rainfall years 2010 (rainfall 830 mm) and 1975 (880 mm). A counter-factual with zero drip irrigation in 2015 is simply not realistic. Borewell depths would have to be well beyond economic drilling. So we had to assume a range from 30-60% the current adoption of drip.

while eucalyptus plantations increased, land fallowing also increased. To separate out the effect of the watershed structures and drip irrigation, we ran the model without them first to quantify their individual effects on recharge or extraction by 2010 (Table 2).

### 6.4 Developing predictive insight by understanding underlying drivers

The multi-scale hydrologic model was only able to reconstruct historical change, by including changing land use, crop-choice, watershed structures, irrigation sagacity and sources of irrigation.

To develop predictive insight, however, the underlying drivers of these changes need to be understood. To interpret these drivers we drew on extensive social science research conducted by the research team in the upper Arkavathy, including a large farmer survey, focus group discussions, in-
terviews and analyses of policy documents. To classify and identify underlying drivers, we relied on previous meta-analyses that found the outcomes in water-scarce coupled human-water systems could be explained by just four sets of factors. (Padowski et al., 2015; Srinivasan et al., 2012).

- Resource: How much water (rainfall, P) is available relative to land and energy (potential evapotranspiration, PET) available and how much aquifer storage is available relative to variability
in P?

- Culture/Economy: What do people living in a region need water for?

- Governance: Are there are enforceable rules to decide who is allowed to take out how much water?

- Technology: Does the society have the capacity to invest in accessing and storing water and
using it efficiently?

This framework can be applied to develop a socio-hydrologic conceptual framework for the upper TG Halli catchment (Fig. 9), that could form the basis of a predictive model in future.



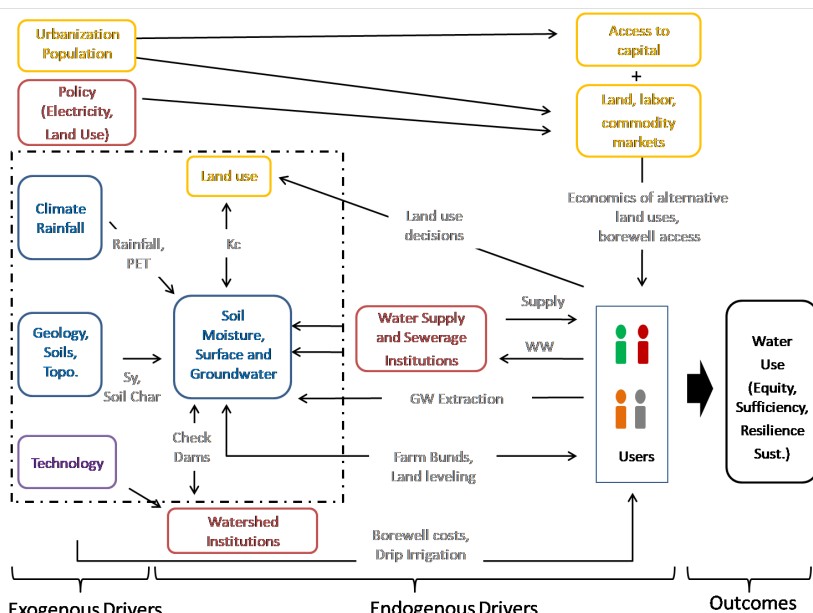

**Figure 9.** Conceptual model of the hydrology of TG Halli watershed. The conceptual model includes both exogenous and endogenous drivers and outcome variables of interest to stakeholders. The factors are color-coded as follows: Blue - Resource, Orange - Economy, Red - Governance, and Purple - Technology. Only the variables in the dotted box, were included in the model presented in this paper. The remaining variables are analyzed either statistically or qualitatively.

### 6.4.1 Resource Drivers

Rainfall in the watershed is highly variable and the landscape water limited (P«PET). Additionally,
the low specific yield (0.1% to 1.5%) of the underlying hard-rock aquifer means that small changes in net extraction result in large changes in average groundwater levels, with large implications for the surface water system (e.g. rapid disconnection of streams from groundwater).

### 6.4.2 Technology Drivers

Borewells were rare in the landscape in the early 70s (Fig S3.1). Interviews with borewell drillers
suggest that rural electrification and introduction of a new "down-the-hole" (DTH) borewell drilling technology spurred the groundwater revolution in the mid-1970s (Balukraya, 2015). The increase in borewell depths is correlated with drop in groundwater levels. As groundwater levels dropped, the number of wells drilled increased (Fig S3.4), suggesting rising drilling costs were not a deterrent. The only price signal on scarcity of water was the substantial cost of drilling a borewell ($4000 to $6000
including the pump and piping costs). The growth of Bengaluru and the markets it provided allowed





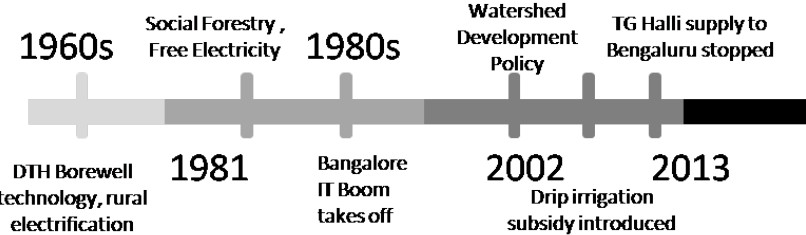

**Figure 10.** Timeline of policy changes in upper Arkavathy watershed

richer farmers diversify income streams to access capital and ultimately drill deeper (Thomas et al., 2015).

### 6.4.3 Governance Drivers

Groundwater has been open access since colonial times. However, post-independence groundwater
pricing and policy was explicitly set with the goals of poverty alleviation and rural development. Neither water nor electricity use for pumping by farmers has been metered since the early 1980s (Fig. 10. The free electricity, open access policy placed no limits on groundwater pumping and allowed rapid rise in borewell drilling.

As surface water disappeared and groundwater dependence increased, traditional systems of gov-
ernance centred around collective management of tanks became irrelevant. A focus group discussion of thirty elderly "neergantis" or village watermen, held in 2015 confirmed that tank irrigation had completely ended by the early 1990s.

Groundwater depletion started receiving policy attention starting in the late 1990s, and spurred watershed development policies to promote artificial recharge through construction of soil and wa-
ter conservation structures, prominently check dams (GOI, 1994, 2002). These were funded under various schemes, implemented by village, district or state level agencies. A field survey of all check dams in the two milli-watersheds indicated that the check dams present today were built after 1995.

### 6.4.4 Economic Drivers

While agriculture was the ostensible driver of hydrologic change, the economic influence of the
rapidly growing city of Bengaluru played an important role in shaping the agricultural and hydrologic systems. Built-up area only accounted for a mere 6.5% of the watershed even in 2014 and direct land conversion to urban uses was clearly not the primary driver of hydrologic change in upper Arkavathy watershed. Instead, the urban influence was enacted by shifting the relative demand and availability of land, labor, and capital.



Multinomial logit analysis (Patil et al., 2017) of crop choice decisions by farmers revealed that in addition to access to groundwater, which is in turn correlated with land holding and wealth, variation in the proximity to the product market and labor availability were important determinants of crop choice. The farm survey data reveal a "quit, diversify or go big" strategy: Farmers either went for a) outright sale of land if they were close enough to the city, b) traditional crops or eucalyptus plan-

tations, supplemented by off-farm income or c) intensification of groundwater irrigated agriculture (Patil et al., 2017) if they had functioning borewells.

Eucalyptus plantations were originally initiated in the region through a World Bank sponsored Social Forestry Programme in the early 1980s that actively promoted eucalyptus, (Shiva et al., 1981). However, contemporary farmer surveys show that today, the choice to plant eucalyptus is driven

entirely by market prices, water and labor scarcity. Labor scarcity was a repeated theme in the focus group discussions. About 75% of the farmers reported it to be a problem and census data confirmed this trend, showing a consistent decline in agricultural employment in the watershed (Fig S2.5 vs. Fig. S2.8). Both labor and water scarcity emerged as statistically significant in choosing eucalyptus (Patil et al., 2017).

## 420  7   Discussion

By conducting a detailed field research campaign we are able to successfully develop sufficient information with which to develop a model to reproduce historical hydrological behavior within the TG Halli watershed. This understanding, not only of hydrological processes but the social drivers of change, create a foundation with which to forecast plausible water futures for the upper Arkavathy.

### 425  7.1   Socio-hydrological narrative of change

We identified three distinct phases in the evolution of the Arkavathy watershed, each characterized by different social controls on the hydrologic system and different hydrologic behavior. The transition between phases was dictated either by technological advances (Phase 1–Phase 2) or socio-hydrologic feedbacks (Phase 2–Phase 3). The drivers of change and the constitutive hydrologic behavior of each

phase is summarized in Figure 11.

#### 7.1.1   Phase 1: Pre-1975. Tank Irrigated Agriculture

Prior to borewells, irrigation was only possible in the command areas of tanks and along streams. The tanks were mainly used to irrigate rice (DES, 1970). In the non-tank command areas, rainfed millets were the primary crops (Fig. S2.1). Groundwater recharge and evapotranspiration from rainfed crops

were the biggest components of the water balance. About 5% to 10% of rainfall ended up as overland runoff (Fig. 11a).



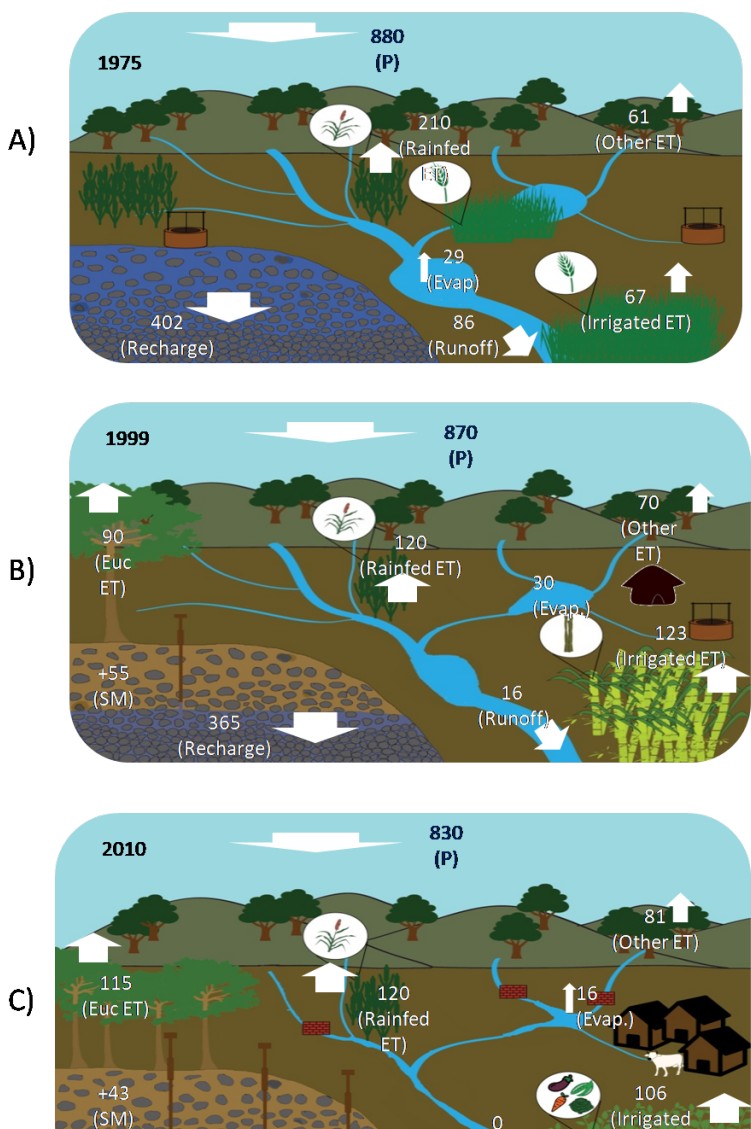

**Figure 11.** Narrative of sociohydrologic change. Shows landscape co-evolution over time





During this period, groundwater, recharged by the streams and tanks, was accessed via shallow open wells. These were located along the stream channels and in the tank command areas as indicated by the distribution of wells drilled before 1975 (Fig. S3.1). Groundwater recharge was far in excess of abstraction in the 1970s. As a result, groundwater levels were shallow as evidenced by open well readings from the period (Srinivasan et al., 2015) and baseflow contributed substantially to streamflows; as much as 20-70%. Tanks filled up and even cascaded during high rainfall events.

### 7.1.2 Phase 2: 1975 to 2000. Shift to groundwater, plantations

Low cost drilling technologies and rural electrification made irrigation more widely available after 1975. As cheap borewell technology became available to farmers, irrigated agriculture increased, resulting in the spatial spread of borewells (Fig. S3.1). Irrigated area peaked during this period (Table 1).

ET from irrigated crops exceeded ET from rainfed crops for the first time (Fig. 11b). As groundwater disappeared and alternative employment opportunities emerged, the shift from rainfed crops to eucalyptus resulted in an increase in actual ET and a decrease in recharge. The combination of low irrigation efficiency and water intensive crops and spread of eucalyptus meant that in volumetric terms, the highest *quantum* of depletion occurred during this period. The shallow weathered aquifer was almost completely depleted and consequently, baseflow disappeared by 1995. By 2000, borewells accounted for virtually all the irrigated area in the two main sub-districts falling within the upper Arkavathy watershed (Fig. S2.7). The well census data indicate that average depth at which water was being struck at over 100 m (Fig. 5). Check dams and other surface storage structures did yet not play a significant role in regulating surface water flows.

### 7.1.3 Phase 3: 2000 to 2015. Commercial agriculture, deep groundwater extraction

Once the shallow aquifer dried, irrigation was only possible with deep borewells. The well census suggests that groundwater levels continued to drop. The average depth at which water was encountered in new borewells, increased from 100 to 200 m over this period (Fig. 5). Borewells spread throughout the milli-watersheds (Fig S3.1) and were now completely unrelated to stream channel locations.

Following government promotion of watershed development, check dams and farm bunds were impounding almost all the remaining runoff by 2015. Although total irrigated area stabilized, the type of crops cultivated also changed. By 2002, rice and sugarcane had completely disappeared. Irrigated land was allocated to "high value" vegetables, coconut, arecanut and orchards. As well yields dropped and well failures increased, farmers drilled even deeper to find deeper fractures. The government introduced a drip irrigation subsidy in 2011. Farmers began to invest in drip irrigation (Fig. 11c).




In villages close to the city, land prices appreciated enough for farmers to sell land to developers and quit agriculture (Lele and Sowmyashree, 2016). Land fallowing in anticipation of sale resulted in a decrease in ET and GW extraction, alleviating water stress.

### 7.2   Policy Insights

This narrative of couple social and hydrologic change provides insights into potential policy responses to ongoing water constraints in the upper Arkavathy. First, in volumetric terms, the worst groundwater depletion occurred when shallow groundwater was still available and irrigation water use was inefficient. This suggests investments in drip irrigation may prove beneficial, provided overall abstraction is kept under control. But more empirical research on actual savings is needed.

Second, implementing watershed development policies without controlling abstraction is clearly ineffective. Watershed development does not create new water; it merely relocates an increasingly insignificant volume of surface water upstream. This suggests a focus on recharge without credible controls on abstraction will not stop groundwater over-exploitation. Third, the study offers both hope and caution. On one hand, the low specific yield of the fracture rock system suggests that re-

saturation of the aquifer maybe possible relatively easily. On the other, even if groundwater levels are improved, because there are thousands of abandoned borewells already in place (Fig. S3.1), farmers will simply pump the groundwater out again, if the current open-access, free electricity policy remains in place.

### 8   Conclusions

The upper Arkavathy watershed has experienced a 80-90% decline in surface and groundwater availability in the past four decades. Secondary analyses showed these declines cannot be attributed to climatic factors, but are almost certainly anthropogenic in origin.

This study reconstructed the decline of ground and surface water in the upper Arkavathy using a multi-scale hydrologic modeling framework that accounted for these anthropogenic effects. The

model was able to attribute the change to eucalyptus, groundwater over-abstraction and soil and water conservation structures.

Analysis of the underlying drivers of change suggests that the system exhibits emergent properties that occur as a result of both endogenous and exogenous drivers. Four sets of human drivers emerged:

- Technological drivers - e.g. cheaper borewell drilling

- Governance drivers - e.g. free electricity, pro-eucalyputs policies

- Urbanization - e.g. access to captial and markets for high value crops

- Endogenous responses to water availability - e.g. check dams, farm bunds, drip irrigation and deeper borewells




Without accounting for these factors, the past four decades of hydrologic change in the TG Halli catchment cannot be understood - let alone can the challenge of predictability into potential futures be confronted. The socio-hydrologic conceptualization presented here offers suggestive insights into how underlying social drivers may shape the water futures of a basin. For example, in urbanizing basins like the Arkavathy, predictions about future water availability must consider the influence of labor and food markets, governance and technology scenarios that influence groundwater abstraction and consequently surface and groundwater availability - in addition to more familiar consideration of the physical setting and future climate trends.

In a system where productivity of the landscape is limited by water, economic drivers will always push for maximization of water abstraction and use. Thus, no natural equilibrium (other than a state of perpetual scarcity) can exist, and instead societies must impose social limits on acceptable levels of water abstraction. In India, successive government policies have focused on increasing efficiency and avoided politically controversial measures to limit consumptive water use. Our study shows that these have had the effect of moving surface water assets upstream and largely to the richest farmers, jeopardizing the well-being of future generations and the poor. Ultimately, only long-term solution is to divide the available renewable resource - rainfall - equitably by ensuring sustainable livelihood patterns that do not exceed the available water resources endowment.



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

*Acknowledgements.* This research was primarily funded under Grant No. 107086-001 from the International
Development Research Centre (IDRC), Canada. Penny acknowledges support from the NSF Graduate Re-
search Fellowship Program under Grant No. DGE 1106400, the NSF and USAID GROW Fellowship Program.
Thompson acknowledges NSF CNIC IIA-1427761 for support of ATREE-UC Berkeley collaborations. Srini-
vasan, Lele and Thomas acknowledge financial support from Tata Trusts. Srinivasan acknowledges support from
the Ministry of Earth Sciences, Government of India under the Newton-Bhabha Fund.There are no conflicts of
interest to the best of our knowledge.

We acknowledge the significant contributions over three years of (late) Kirubaharan Jeremiah. We thank
Apoorva R. for active support of installing maintaining the field instrumentation. We thank the field hydrol-
ogy and support teams including Manjunatha, Chidanand, Janardhan, Usha, dozens of interns and community
volunteers. We thank Muneeswaran, Sowmy and others from the ATREE Ecoinformatics Lab for their RS/GIS
mapping work. We acknowledge data from the following government agencies: Central Ground Water Board,
Karnataka State Drought Monitoring Centre and KA Department of Economics and Statistics. We are grateful
to the Hadonahalli and Aralumallige Gram Panchayats (village councils) for allocating land to set up weather
stations. We thank colleagues from the socio-hydrology community for their critical comments that helped
improve this study over the years. We are, however, responsible for any remaining errors.