# Peer review of "Proximate and underlying drivers of socio-hydrologic change in the upper Arkavathy watershed, India"

_Hydrology and Earth System Sciences, 2017_

## Referee Comment (RC1) · Anonymous Referee #1 · 5 Oct 2017

The paper by Srinivasan et al. present a study to define the drivers of socio-hydrological change in an Indian river basin. In the paper, both a qualitative narrative of the development of the basin, as results from a modeling exercise are presented. Although the idea of the paper is very relevant and suitable for HESS, I believe that substantial additional work is necessary before publication in HESS can be considered. I find the paper written in a very unstructured way, and it is not very clear what the motivation and approach of the research are. In the following, I motivate my major concerns, followed by some minor comments. At its current state, I suggest rejection of the paper. However, I do encourage the authors to substantially rewrite the paper for resubmission to HESS.

Major concerns

1. In general the paper reads unstructured. It is not clear what to expect, and the research methods, results and discussion are presented in a very on-the-fly fashion. I believe the paper can be rewritten in a more concise, clear, and structured way. The research questions appear in different forms throughout the manuscript, which are in itself poorly defined (see my comments later). It is not clear what the scientific merit of this work is, and what the greater socio-hydrological community can learn from this study.

2. Introduction: It is not clear from the introduction what the study is about. The authors illustrate that there's a need for hydrological models taking into account 'change', but the motivation to study the Arkavathy basin is unclear. Also, I feel that that the connection to existing socio-hydrological literature (on previous modeling efforts specifically) is missing. I suggest that the authors cut down on the amount of text illustrating how "urgent" it is to take into account change etc., and give a better introduction of the study area, the motivation and the taken approach.

3. Model: Please elaborate on why this modeling approach was chosen. Why was a multi-scale model used? It seems like the model comes with relatively high complexity, while one of the issues is data scarcity. Why did the authors not design a simple conceptual model? What makes this case study unique that no previously developed socio-hydrological modeling approaches could be applied here? Please also elaborate on the choice of different time-scales of the different model components. Why was the model not completely run at the lowest time scale? And why is the later presented socio-hydrological framework not used as a modeling framework

4. Discussion: The narrative is very interesting and key to understand the system. However, I would propose to include this before the modeling exercise. By first thoroughly introducing the basin, and understand qualitatively what happened, the modeling exercise and quantitative analysis follows easier (e.g. similar to the work in the

Murrumbidgee River Basin, first the qualitative analysis by Kandasamy et al. (2013) followed by the modeling work by van Emmerik et al. (2014)), see Mostert (2017).

5. I think the paper also misses a coupling to the socio-hydrological work in the recent years. Only few references are made to other modeling papers, and it seems like no socio-hydrologic literature from the past years is really used to develop a sound research method.

6. Quality of the figures should be improved considerably. The quality of a paper is, in my vision, reflected in the quality of the figures. The resolution is low, the font type can be more professional, and the titles and axis are sometimes inconsistent.

Minor comments

1. L28: Hydrologic non-stationarity?

2. L38: What is mean with rapid? Why is it italic?

3. L40: Rapid what?

4. L36-42: I don't really see the point of this paragraph, there's some oneliners, that doesn't necessary add to an argument.

5. L43-50: I suggest to summarize this paragraph in 1-2 sentences, and add it to L28-35.

6. L51-54: Rewrite, introduce proximate vs underlying drives, and then use land-use as an example.

7. L60: Sure, hydrologic models are not equipped to diagnose changes, but socio-hydrological models do. Quite some recent efforts on case studies (e.g. Elshafei et al., 2014; van Emmerik et al., 2014; Lui et al., 2015; Chen et al., 2016; M) developed models to study the influence of these changes. It would be good if the authors briefly reflect on these advances, and specify what

8. L83: Give a better introduction of the term "tanks".

9. L106: The authors can better remind the reader of what was found in the previous study, and what unknowns and questions were identified.

10. L107: Question can be posed better. What is meant with change? What proximate drivers are meant? All of them? Several specific ones?

11. L108: What is meant with "we"? We as a human race? Hydrologists? Policy makers? Water managers in the study area? What is meant here with change? "These underlying drivers" refer to the proximate drivers from question 1 or to actual underlying drivers? What is meant with predictive insights? Insights in river discharge? Evaporation? Water use?

12. L110: Implications for who? Rephrase question.

13. L120: Why is theory-based in parentheses?

14. L130: Year missing in Penny et al. (throughout the paper).

15. L130: Good to see that the authors found a solution to minimize equifinality, but how was this done for this study? One sentence paragraphs also read a bit awkward, so I suggest merging with the previous paragraph or expanding.

16. L132: Fig. 2 is introduced, but not explained. This model is rather crucial for the rest of the study, so I suggest to elaborate on it further.

17. L134: What is meant with "the data"?

18. L142: Why is the term "milli-watersheds" used? What does it mean?

19. L165-166: Provide some more details of these measurements as they were used to identify the dominant runoff mechanisms. How often were these measurements taken? How were they used to identify the dominant mechanisms?

20. L173: Please elaborate on the crowd-sourced data. Not clear what these are.

21. L202-204: The rest of the plots are still flood irrigated?

22. L280-308: This section reads very confusing. I expect to read about the calibration strategy, and what data was used etc. The authors however describe and interpret results here.

23. L310: Please first introduce the results before concluding it was able to replicate the data.

24. L312: How did the simulation revealed the relative importance of different drivers?

25. L350 -367: Very messy paragraph. Reads like it's still work in progress. Are the questions the framework? If so, how are the questions answered?

26. L367: Why is the socio-hydrological framework not used as a modeling framework?

27. Go through all semi-colons and assess whether it's really necessary to use them.

References

Chen, Xi, et al. "From channelization to restoration: Sociohydrologic modeling with changing community preferences in the Kissimmee River Basin, Florida." Water Resources Research 52.2 (2016): 1227-1244.

Elshafei, Y., et al. "A prototype framework for models of socio-hydrology: identification of key feedback loops and parameterisation approach." Hydrology and Earth System Sciences 18.6 (2014): 2141-2166.

Kandasamy, J. K., et al. "Socio-hydrologic drivers of the pendulum swing between agricultural development and environmental health: a case study from Murrumbidgee River basin, Australia." Hydrology and Earth System Sciences (2014).

Liu, Dengfeng, et al. "A conceptual socio-hydrological model of the co-evolution of humans and water: case study of the Tarim River basin, western China." Hydrology and Earth System Sciences 19.2 (2015): 1035-1054.

[Figure]

Mostert, E.: An alternative approach for socio-hydrology: case study research, Hydrol. Earth Syst. Sci. Discuss., https://doi.org/10.5194/hess-2017-299, in review, 2017.

Roobavannan, M., et al. "Allocating Environmental Water and Impact on Basin Unemployment: Role of A Diversified Economy." Ecological Economics 136 (2017): 178-188.

Van Emmerik, T. H. M., et al. "Socio-hydrologic modeling to understand and mediate the competition for water between agriculture development and environmental health: Murrumbidgee River basin, Australia." Hydrology and Earth System Sciences 18.10 (2014): 4239.

---

## Referee Comment (RC2) · H. McMillan (Referee) · 19 Oct 2017

This paper describes a large, integrated study of the upper Arkavathy watershed in India, where anthropogenic change has caused substantial declines in groundwater and reservoir levels. The study employs sociological and hydrological modelling methods in an attempt to determine the dominant causes of these declines. The paper is generally well written and interesting to read, and the subject of the study fits well into the scope of HESS.

Major comments:

[Figure]

1. The structure of the paper is confusing for the reader. On reading, it appears as two papers back to back – the first addressing building and evaluating the nested hydrologic model (up to Section 6.3), and the second an investigation of the sociological drivers (Sections 6.4 and 7). Currently there is little connection between the two. I think either the second part should either be removed to a different paper, or it should be included up front as part of literature review and model development, and then the authors would need to show how this information is used within the hydrologic model.

Other comments:

1. Several typos throughout the manuscript, please proofread.

2. Line183 - The reasons for discounting the monitoring well data are not very well motivated, in such a data scarce catchment surely it adds some information?

3. In the methods section it is not always clear which work was already completed as part of previous studies in the watershed, and what is new for this paper. It would be helpful if the authors can try to clarify this where possible.

4. Line255 - The assumption of no groundwater connectivity between tank aquifers does not seem realistic even if there are no large fractures. There doesn't seem to be any reason why groundwater would be connected within tank basins and not connected outside. The authors should at least discuss the limitations of this assumption.

5. Figure 5. Please clarify in the caption whether this is data or model output.

6. The MWF model seems as though it would be very sensitive to rainfall intensity, but the rainfall is downscaled data and so may not represent the intensity accurately over large areas. Please can the authors comment on what impact this could have on the model accuracy.

7. Figure 7. From looking at the figure, recharge seems to be defined as "water below rooting depth" meaning that the depth at which water is judged to become recharge depends on the crop. Is this correct, and if so shouldn't recharge be deemed to begin
at a consistent depth?

8. Line322. Do you mean that groundwater decline and land use change cannot explain runoff decline under any circumstances, or just that it did not work in your model?

9. Some conclusion is needed at the end of the modelling section. Was the model deemed to be good/bad/useful? How will it be used in future? This is partly due to the problem of the paper structure, as normally the paper discussion and conclusion would be sited here to discuss the success or otherwise of the modelling effort. I would also suggest that the information in Supplement 5 be added here as part of the model discussion.

10. Table 2. The table seems to show more causes of groundwater increase than groundwater decrease, does this mean that groundwater should be increasing?

---

## Referee Comment (RC3) · C. Scott (Referee) · 30 Oct 2017

This manuscript synthesizes new conceptual thinking, extensive empirical data collection, socio-hydrologic model development, consideration of drivers of changes, and makes well-substantiated conclusions.

I am attaching an annotated version of the manuscript with detailed comments. My regrets for the delay beyond the requested date for referee comments.

What makes this paper unique in the emerging field of socio-hydrology is the rigorous testing of hypotheses using meticulously collected multi-methods data.

[Figure]

I believe the governance and policy discussions could be strengthened; however, for HESS readership, I consider these sections to be appropriated.

Please also note the supplement to this comment:
https://www.hydrol-earth-syst-sci-discuss.net/hess-2017-543/hess-2017-543-RC3-supplement.pdf

**Supplement:**

Manuscript prepared for Hydrol. Earth Syst. Sci.
with version 2014/09/16 7.15 Copernicus papers of the LATEX class copernicus.cls.
Date: 6 September 2017

[revised manuscript text omitted]

---

## Author Comment (AC1) · 18 Mar 2018

The reviewer made several substantive points including recommendations to restructure the paper and to adopt a stylistic sociohydrologic modeling approach. We fully accept the structural critique and will substantially restructure the revised paper. However, we respectfully disagree with the reviewer's suggested sociohydrologic modeling approach. The reasons are somewhat subtle.

Firstly we agree that fully incorporating bi-directional feedbacks between human agent models and hydrologic models in either a detailed framework based on primary studies or simple conceptual model would be intriguing in the Arkavathy Basin. Developing

such a model remains important future work. We further agree that existing sociohydrologic studies should be more extensively cited in our revision to motivate the present work. That being said, the goal of the paper is not to immediately present such a socio-hydrologic model, but instead to use a traditional hydrologic modeling approach to illuminate the lack of predictive power of such approaches in the human-dominated environment we explore.

We believe that such a demonstration is important for several reasons: 1. Motivating socio-hydrology: Socio-hydrology remains a fringes area of study amongst the water science community in India and beyond. Providing a clear illustration of the necessity for a sociohydrologic modeling framework therefore remains a necessary scientific task in this intellectual environment. Our approach is to demonstrate that models that explicitly incorporate human feedbacks are necessary to have any predictive ability. The goal of this study is to disentangle the role of human factors in explaining long term change – thus motivating the need for socio-hydrology.

2. Illuminating the importance of human factors for interrogating periods of hydrological change: A widely used approach to exploring basin-scale hydrology remains the calibration of large basin-scale hydrologic models using secondary data. These models are difficult to adapt to situations where human interventions generate changing rainfall-runoff relationships. As a solution of last resort, some hydrologists are even now making the case that model parameters should evolve over time to enable improved model calibration. Such approach makes it possible to reconstruct hydrology. However it does not provide insight into why watershed parameters are changing and offers no predictive insight or organizing principles. Our paper offers an alternative approach that prioritizes the development of process understanding over model performance metrics.

3. Informing policy: Much of the hydrologic change occurring in India can be attributed to the cumulative impacts of millions of uncoordinated actions by humans. "Watershed development" (constructing soil and water conservation structures) is the cornerstone

of both Indian water policy and rural development. In these programmes, the impacts are always measured locally; the cumulative impacts of local water harvesting at the basin scales are never understood resulting in increasing upstream downstream conflict. Multi-scale models are necessary to understand how small scale interventions scale up. It is clear that the current presentation of the paper needs to be amended to motivate the research with these challenges and to demonstrate the value the current modeling approach offers in this context. We agree with Reviewer 2 that the paper attempts to present too much information and needs to be streamlined and focused. Our proposed revision will substantially restructure the paper and simplify its argument. A skeleton outline is presented below: 1. Introduction. Human impacts are the primary drivers of change. Review the literature on how others are dealing with human drivers – either toy models, or small scale models or allowing model parameters to evolve in basin scale models. Toy models are useful to understand the broad dynamic and direction of change but not for quantitative reconstruction.

2. Conceptual Model We will then present a narrative of change drivers and a conceptual model and explain why this necessitates a multi-scale model – to accommodate millions of small changes that occur at different scales.

3. Results We will then present the results of the model – and show the trajectory of change and attribution to the main drivers.

4. Discussion We will discuss the results in terms of attribution of causes and introduce the idea of urbanization as an underlying driver and the challenges this poses for prediction.

5. Conclusion We will conclude with big picture implications for both hydrologic science and policy.

Minor points: Comment Response The narrative is very interesting and key to understand the system. However, I would propose to include this before the modelling exercise. The proposed restructuring of the paper accommodates this request.

Quality of the figures should be improved considerably OK. We will hire a professional to do this.

The reviewer offered a number of very specific edits, which will likely be obviated by the proposed restructure. Where appropriate in the restructured manuscript, these will be retained. We thank the reviewer for their helpful and careful feedback.

Please also note the supplement to this comment:
https://www.hydrol-earth-syst-sci-discuss.net/hess-2017-543/hess-2017-543-AC1-supplement.pdf

---

## Author Comment (AC2) · 18 Mar 2018

**Reviewer 2:**

Reviewer 2, Hilary Mcmillan, has been largely supportive but has requested a substantial revision of the paper. We thank the reviewer for her careful reading. The minor changes will be incorporated in the rewriting.

| Comment | Response |
|---|---|
| The structure of the paper is confusing for the reader. On reading, it appears as two papers back to back – the first addressing building and evaluating the nested hydrologic model (up to Section 6.3), and the second an investigation of the sociological drivers (Sections 6.4 and 7). Currently there is little connection between the two. I think either the second part should either be removed to a different paper, or it should be included up front as part of literature review and model development, and then the authors would need to show how this information is used within the hydrologic model. | We agree and thank the reviewer for helping us think this through. Several reviewers have raised this issue. We believe that the proposed restructuring of the paper will address this. |
| Figure 7. From looking at the figure, recharge seems to be defined as "water below rooting depth" meaning that the depth at which water is judged to become recharge depends on the crop. Is this correct, and if so shouldn't recharge be deemed to begin at a consistent depth? | The reviewer is correct that we make the assumption that water that travels below the rooting depth of the crops is available to recharge. This reflects the low topographic gradients in the catchment and the assumption that lateral flow is negligible except in close proximity to the land surface. In the absence of lateral flow, a one-dimensional water budget is appropriate, and recharge will represent any water that cannot be utilized by vegetation. |
| Line 255 - The assumption of no groundwater connectivity between tank aquifers does not seem realistic even if there are no large fractures. There doesn't seem to be any reason why groundwater would be connected within tank basins and not connected outside. The authors should at least discuss the limitations of this assumption. | Analysis of fracture networks and well-to-well connectivity suggests that there is minimal lateral connection between wells even on distances of 5-10m. Thus the issue is not that lateral flow between tank aquifers is neglected, but rather that we aggregate and thus "average" storage across the fine-grained variations in the field to the tank scale. This raises the possibility that nonlinear interactions between local water storage and water use that could amplify these effects are being inappropriately averaged. We do not, in fact, introduce significant nonlinear assumptions regarding storage, so the averaging should introduce little error. This will be discussed in the revised paper. |
| The MWF model seems as though it would be very sensitive to rainfall intensity, but | MWF itself is not sensitive to intensity (i.e. the development of wetting fronts is insensitive to |

| | |
|---|---|
| the rainfall is downscaled data and so may not represent the intensity accurately over large areas. Please can the authors comment on what impact this could have on the model accuracy. | intensity). However the input to MWF – the infiltration flux, is sensitive to intensity - as any model that accounts for land surface partitioning of rainfall must be. Specifically, it is the relationship of intensity of the infiltration rate (Ksat) that is important. Both rainfall and Ksat have significant uncertainty in them – and Ksat is ultimately calibrated, subject to the available rainfall intensity data. Thus, improved rainfall downscaling might give a more certain estimate of Ksat, but given the calibration process, the water budget impacts would be minimal.

Ultimately, the main point of the paper is to explain the long term decline in surface runoff. We have already established that there are no trends in rainfall intensity through multiple rain gages. Thus, we do not anticipate that the accurary of model results would change significantly given improved rainfall downscaling. |
| Do you mean that groundwater decline and land use change cannot explain runoff decline under any circumstances, or just that it did not work in your model? | Groundwater decline and land use change explains the stream aquifer disconnection and consequent baseflow decline (all of which occurred by 1995), as has been documented by many studies world over.

After the mid-1990s, the link between stream flow decline and groundwater depletion is not as direct.
There is no hydrologic (ie non-anthropogenic) mechanism that we believe could explain the continued decline of surface runoff beyond the mid-1990s.
This is not an artefact of our model, but involves mechanisms common to all watersheds in India, which have been heavily modified with watershed structures. |
| Some conclusion is needed at the end of the modelling section. Was the model deemed to be good/bad/useful? How will it be used in future? This is partly due to the problem of the paper structure, as normally the paper discussion and conclusion would be sited here to discuss the success or otherwise of the modelling effort. I would | We agree. We will address this in the revised and restructure manuscript. |

| | |
|---|---|
| also suggest that the information in Supplement 5 be added here as part of the model discussion. | |
| Table 2. The table seems to show more causes of groundwater increase than groundwater decrease, does this mean that groundwater should be increasing? | Yes. But the relative magnitudes of these contributions also matter. We will clarify this in the revision. |

---

## Author Comment (AC3) · 18 Mar 2018

**Reviewer 3:**

Reviewer 3, Christopher Scott, has been largely supportive and has only asked for technical revisions. offered a number of specific comments including wording and punctuation changes. We thank the reviewer for his careful reading. These will be incorporated in the rewriting. In this response, we only address the substantive comments.

| Comment | Response |
| --- | --- |
| This manuscript synthesizes new conceptual thinking, extensive empirical data collection, socio-hydrologic model development, consideration of drivers of changes, and makes well-substantiated conclusions. I am attaching an annotated version of the manuscript with detailed comments. My regrets for the delay beyond the requested date for referee comments. What makes this paper unique in the emerging field of socio-hydrology is the rigorous testing of hypotheses using meticulously collected multi-methods data. | We thank the reviewer for his supportive comments. |
| qualify this. entirely flood irrigation in 70s to what mix presently?. efficiency increases with area expansion contribute to water-resource depletion | These figures will be added |
| need sensitivity analysis. | Since we are now shortening the latter half of the paper, we will be adding sensitivity analyses in. |
| while I agree, your analysis has not demonstrated this | We propose to remove this from this paper and keeping only the main hydrology argument. |

---

## Editor Comment (EC1) · M. Sivapalan (Editor) · 26 Mar 2018

The reviewers raised serious concerns about the paper, and especially about how the paper was structured. I agree with their comments 100%.

There is material here that is very interesting and very relevant to the new field of socio-hydrology, and for this reason I am willing to allow a resubmission of the paper along the lines that the authors have indicated.

Because of the substantial changes that the reviewers requested and the authors have agreed to do, it has to go through a thorough re-review, including possibly 2 of the old

reviewers.

I look forward to this resubmission, which I will undertake to get reviewed as quickly as possible.

─────────────────────────